# Enhancing Methane Production through Anaerobic Co-Digestion of Sewage Sludge: A Modified ADM1 Model Approach

**Khuthadzo E. Mudzanani [1], Terence T. Phadi [1], Sunny E. Iyuke [2] and Michael O. Daramola [2,3,***

[1] Measurements and Control, Mintek, Praegville, Randburg 2194, South Africa
[2] School of Chemical and Metallurgical Engineering, Faculty of Engineering and the Built Environment, University of the Witwatersrand, Johannesburg 2050, South Africa
[3] Department of Chemical Engineering, Faculty of Engineering, Built Environment and Information Technology, University of Pretoria, Hatfield, Pretoria 0028, South Africa
* Correspondence: michael.daramola@up.ac.za

**Abstract:** The International Water Association's (IWA) established Anaerobic Digestion Model No. 1 (ADM1) was created to serve as a backup for experimental findings regarding the actual anaerobic digestion process. The previous model idea was adjusted and used to simulate an anaerobic digestion process in this study. Testing procedures, such as benchmark tests and balance checks, were performed in order to verify the accuracy of the implementation. These measures worked in tandem to ensure that the model was implemented flawlessly and without inconsistencies. The primary objective of this article is to construct a method that is based on the ADM1 for evaluating co-digestion and predicting the performance of the digestion process or methane yield based on the analyzed substrates' physicochemical properties. Additional equations and simulations have been added to the standard model to create tools for evaluating the feasibility of anaerobic co-digestion. The study's two most intriguing aspects are the optimal mixture and parameter dependence. The adjusted ADM1 is accurate in predicting the measured values of effluent COD, pH, methane, and produced biogas flows with a reasonable degree of accuracy, according to the validation results. This research shows how to use ADM1 in a wastewater treatment plant and other settings where anaerobic digestion is of interest.

**Keywords:** Anaerobic Digestion Model No. 1 (ADM1); MATLAB simulation; anaerobic co-digestion; sewage sludge; dairy waste

## 1. Introduction

Anaerobic digestion is a well-known biological treatment process that results in the production of valuable methane gas through the degradation and stabilization of municipal sewage sludge [1–3]. The need to reduce operating costs, as well as changing standards for the use and disposal of sewage sludge, has sparked interest in more effective sewage sludge treatment processes [4]. One method for increasing efficiency is to physically divide the single-stage anaerobic digestion process into two process stages: a thermophilic pre-treatment stage followed by a mesophilic main treatment stage [5].

The advancement of a different mathematical model for a variety of substrates has resulted from the growing interest in anaerobic digestion modeling in recent years. The Anaerobic Digestion Model 1 (ADM1) was published in the IWA's Scientific and Technical Report No. 9 and has since become the most relevant model developed for the simulation of anaerobic treatment [6,7]. The ADM1 was designed in such a way that a standard parameter, Chemical Oxygen Demand (COD), and the process configuration of a continuously stirred tank reactor were used to determine the composition of various wastes [8].

The biochemical activities covered in the ADM1 are classified as composite disintegration, substrate degradation mechanisms, hydrolysis of particulate COD, and particular biomass growth and decay processes (see Figure 1). In complex particulate waste, carbohydrate, protein, and lipid particle substrates, as well as particulate and soluble inert material, dissolve first. The disintegration step was largely implemented to facilitate the modeling of activated sludge digestion. This allows for the lysis of biological sludge and complicated organic matter, with the composites acting as a pre-lysis store for degrading biomass [9].

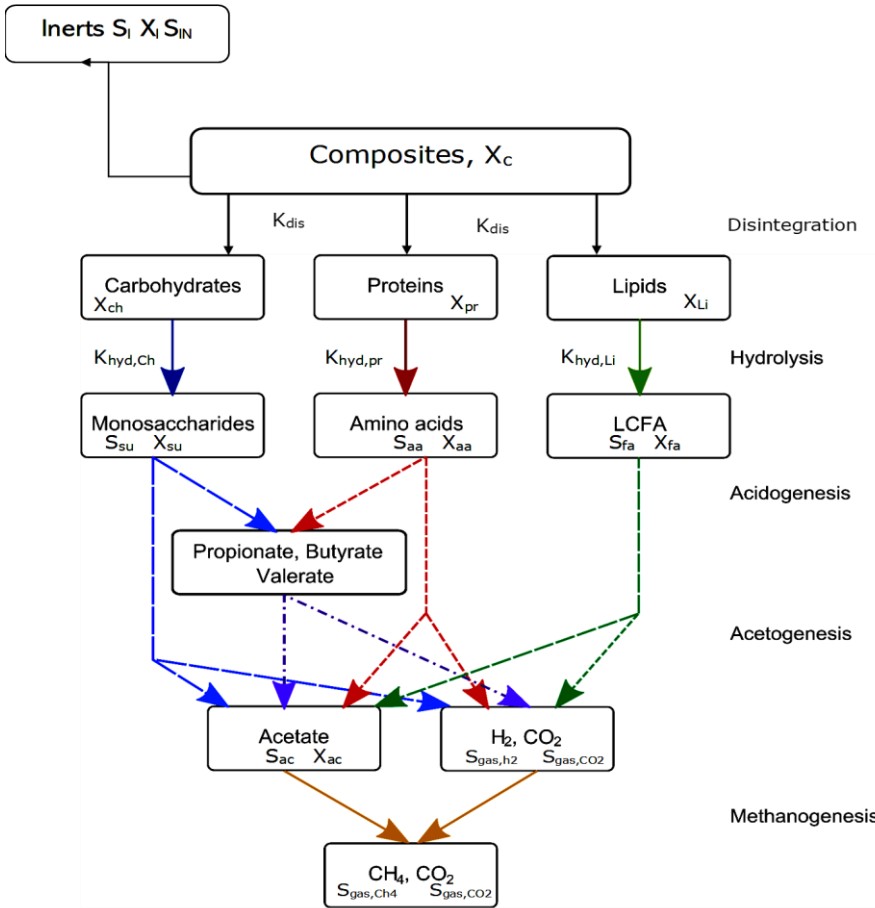

**Figure 1.** Anaerobic digestion process—composites degrade to inerts and hydrolysis reactants through a disintegrating step.

The enzymes degrade proteins, carbohydrates, and lipids into monosaccharides, amino acids, long-chain fatty acids (LCFAs), and other molecules, which are subsequently hydrolyzed into monosaccharides, amino acids, LCFAs, and other molecules, see Figure 1. Monosaccharides and amino acids are absorbed and converted into VFAs and hydrogen during the acidogenesis step [10,11]. Molecular hydrogen and acetate are produced via anaerobic LCFA oxidation. In the acetogenesis stage, propionate, valerate, and butyrate are decomposed to produce molecular hydrogen and acetate. Methane is produced via both aceticlastic methanogenesis (the cleavage of acetate to methane) and hydrogenotrophic methanogenesis (the reduction of carbon dioxide to methane) [10,12].

Similar to ADM1, Anaerobic Digestion Model No. 2 (ADM2) is an advanced mathematical model that describes the complex biochemical processes in anaerobic digesters, primarily for treating organic materials and producing biogas, primarily containing methane [13]. ADM2 is a valuable tool for understanding and optimizing anaerobic digestion processes and accommodates a wider range of substrates. It incorporates more microbial groups and interactions and employs more detailed kinetic equations [14]. However, ADM2 is more complex than its predecessor ADM1, which means it requires a larger number of

parameters to be estimated. Parameter estimation can be challenging and time-consuming, as obtaining accurate parameter values often requires extensive experimental data [14]. The increased complexity of ADM2 compared to ADM1 can make simulations computationally intensive, particularly for larger and more intricate systems. This can be a limitation when trying to model real-time or large-scale processes [15]. The model represents microbial growth and metabolic kinetics using mathematical equations [15]. However, microbial activity can vary significantly due to factors such as pH, temperature, and changes in substrate composition, making it more complicated to modify to suit a specific application, such as co-digestion; hence, ADM1 was used for this study [14,16,17].

ADM1 has its own disadvantages, including parameter uncertainty, model calibration and validation, sensitivity to initial conditions, limited representation of microbial diversity, inadequate handling of process dynamics, incomplete kinetics, and lack of specificity for different feedstocks [18]. Complexity in ADM1 involves numerous equations and parameters, making it challenging to parameterize, calibrate, and use effectively. Parameter uncertainty can lead to significant uncertainties in model predictions, and model calibration and validation require high-quality experimental data [19]. Sensitivity to initial conditions and parameter values can also limit the model's reliability and applicability. It often has a limited representation of microbial diversity, inadequate handling of process dynamics, and inability to incorporate external factors that can limit the model's accuracy in predicting metabolic activities [20]. Researchers continuously work on refining ADM1 and developing more advanced models to better capture the complexity of anaerobic digestion processes while considering diverse operational and environmental conditions.

The Modified ADM1 Model is an adaptation of the original ADM1 model to address the anaerobic co-digestion of sewage sludge for methane production [21]. The Modified ADM1 Model includes core components of the original ADM1, including biochemical reactions, microbial populations, and parameters describing substrate degradation, biomass growth, and biogas production [22]. The model incorporates the specific characteristics and composition of sewage sludge and other co-substrates used in the co-digestion process, including organic matter content, nutrient composition, and inhibitory substances that might affect the digestion process. It accounts for the interactions between different microbial groups involved in the co-digestion process, such as breaking down complex organic compounds into simpler compounds and producing methane [23,24].

Kinetic parameters in the Modified ADM1 Model are adjusted to reflect the specific behavior of microorganisms in the co-digestion process, such as determining the rate at which different substrates are degraded and converted into biogas [25]. The model also accounts for inhibition effects, such as heavy metals or toxic compounds, if present in the sewage sludge or co-substrates [26]. The Modified ADM1 Model can be used to predict the performance of the anaerobic co-digestion process, estimating the amount of biogas (methane) produced, the rate of digestion, and the dynamics of microbial populations over time. It can also be utilized for process optimization, allowing operators and researchers to explore different scenarios to enhance biogas production efficiency and overall process performance [27,28].

The Modified ADM1 Model applied to the anaerobic co-digestion of sewage sludge for methane production serves as a valuable tool for understanding and optimizing the complex interactions and processes involved in waste-to-energy technology. The modified ADM1 model accounts for the co-digestion of sewage sludge with organic substrates, introducing greater complexity and complexity [29]. This model can simulate interactions between different substrates and their impact on methane production. The model's incorporation of multiple substrates improves prediction accuracy, process optimization, and biogas composition. It also predicts changes in biogas composition due to co-digestion, which is crucial for energy generation quality [30]. The model is flexible for research and design, allowing researchers and engineers to explore different scenarios and substrates without trial-and-error experimentation. It also allows for environmental and economic impact assessment, helping decision-makers understand the potential benefits of anaerobic

co-digestion compared to traditional single-substrate digestion. The modified ADM1 model can be adapted to accommodate advancements in anaerobic digestion technologies and the inclusion of new substrates, making it a versatile tool for long-term planning [31].

The study focuses on defining the biodegradation kinetics of anaerobic digestion using the ADM1 simulation approach. This mathematical framework describes biological, chemical, and physical processes during anaerobic digestion, considering parameters like substrate composition, reaction kinetics, and microbial interactions. The primary objective is to estimate the performance of anaerobic digestion, predicting outcomes like methane production, biogas composition, and substrate degradation rates. The study focuses on sewage sludge, a by-product of wastewater treatment, and aims to define the biodegradation kinetics of sewage sludge during anaerobic digestion. Parameterization and calibration are crucial for accurate simulation and performance estimation. The results could have implications for wastewater treatment plants and anaerobic digestion facilities, influencing process optimization, reactor design, substrate selection, and overall efficiency improvement. Biogas production and energy generation are also potential benefits, as methane-rich biogas is a main by-product of anaerobic digestion. The study contributes to the field of anaerobic digestion by refining our understanding of the biodegradation kinetics of sewage sludge and advancing our knowledge of organic compounds degrading in anaerobic conditions.

## 2. Materials and Methods

### 2.1. Substrate and Inoculum

The sewage sludge was collected at random from municipal WWTWs in Gauteng Province, South Africa, over six months. The manure was collected from the Cavalier abattoir in Pretoria, Gauteng, South Africa. The sampling points were carefully selected, as they are representative of the entire flow's cross-section, resulting in a well-mixed sample. A grab sampling technique from anaerobic digestion influent streams was used as the standard sampling method. Prior to analysis, the samples were kept in a cold room inside a 4 °C laboratory fridge. On the AD feedstock, samples were set aside for metagenomics analysis and physicochemical tests.

### 2.2. Batch and Semi-Continuous Tests

The collected substrates were tested in batches to obtain the kinetic constants and assess their mechanism. Since the substrates were gathered over a lengthy span of time, batch studies were also used to verify the activity fluctuation of the substrates. More information on the batch and semi-continuous study arrangement can be found in Mudzanani et al. [4].

### 2.3. Analytical Method

The VS, TS (VSS, TSS), TA, TKN, and concentrations of N—NH$_3$, N—NO$_2$, and N—NO$_3$ were determined using Standard Methods (APHA, 2012); COD was determined using Cell Tests on samples centrifuged (10 min at 4000 rpm) and filtered at 0.45 μm (MERCK-referring to EPA 410.4 method). Based on the total COD of the sewage sludge, organic matter was fractioned into 40% inert particulate COD, 30% hydrolysis products, and 30% particulate degradable COD. These were further divided into 9% amino acids, 6% sugars, 13.5% LCFA, and 1.5% inert soluble COD [20]. The inputs that were applied to the different factors of the constructed model are listed in Table 1.

**Table 1.** Measured parameters as inputs for ADM1.

| Component | Description | Min | Max | Average |
|---|---|---|---|---|
| Flow ($m^3$/day) | Influent Flow | 165 | 170 | 167.5 |
| pH | | 5.1 | 7 | 6.05 |
| Temperature (°C) | | 31 | 37 | 34 |
| TCOD (mg/L) | Total COD | 7940 | 10,500 | 9220 |
| SCOD (mg/L) | Soluble COD | 5342 | 8500 | 6921 |
| PCOD (mg/L) | Particulate COD | 2598 | 1000 | 1799 |
| TSS (mg/L) | Total Suspended Solids | 603.44 | 750.03 | 676.735 |
| VSS (mg/L) | Volatile Suspended Solids | 573.268 | 609.23 | 591.249 |
| VFA (meq/L) | Volatile Fatty Acids | 721.3 | 821.56 | 771.43 |
| Total Alkalinity | as $CaCO_3$ | 584 | 589 | 586.5 |
| TOC | Total Organic Carbon | 25 | 31 | 28 |
| TKN (mg N/L) | Total Kjeldahl Nitrogen | 925 | 946.3 | 935.65 |
| $NH_4$-N (mg N/L) | Ammonium Nitrogen | 127 | 129 | 128 |
| $NO_2$-N (mg N/L) | Nitrate | 54 | 54.3 | 54.15 |
| $NO_3$-N (mg N/L) | Nitrite | 190 | 197 | 193.5 |
| Total-P (mg P/L) | Total Phosphorus | 3.69 | 6 | 4.845 |
| $PO_4$-P (mg P/L) | Orthophosphates | 0.1 | 0.12 | 0.11 |
| Ca (mg/L) | Calcium | 5.4 | 5.4 | 5.4 |
| Mg (mg/L) | Magnesium | 0.87 | 0.87 | 0.87 |
| $SO_4$ (mg/L) | Sulphates | 0.32 | 0.45 | 0.385 |
| $CH_4$ content (%) | Methane % | 65 | 71 | 68 |
| $CO_2$ content (%) | Carbon dioxide | 31 | 32 | 31.5 |

*2.4. Mathematical Model Development*

Mathematically, ADM1 is described as a non-linear structured ODE model that depicts the anaerobic digestion reaction in a CSTR by incorporating several biological and physicochemical phenomena [32]. In this process, biodegradable organic compounds are transformed into carbon dioxide and methane with a trace of inert by-products. The key biochemical processes studied start with the disintegration stage, followed by the hydrolysis step, acidogenesis, and acetogenesis until methanogenesis, refer back to Figure 1.

2.4.1. Model Structure Philosophy

International collaboration on anaerobic process technology developed the ADM1 model, aiming to overcome the limitations of previous models due to their specialized nature. This ADM1 generally has biochemical and physical mechanisms that are widely accepted, similar to those applied in Alzheimer's research [33]. The Petersen matrix, which describes stoichiometric conversions and kinetic rates, has the same format. ADM1's general framework allows for uniform representation of precipitation, complexation, and adsorption models. To introduce the basic methodology of modeling conceptualized ADM1, a simple representation of the soluble component, I ($S_i$), particulates ($X_i$), and gas component ($S_{GAS}$) is presented. The model was built by first defining the various reactions that will be considered. As shown in Figure 2, substrates are taken up by cells in a particulates-containing liquid influent stream [7]. The AD reaction takes place, converting the substrate to products (gas phase) and various biomass constituents (liquid and particulates phase).

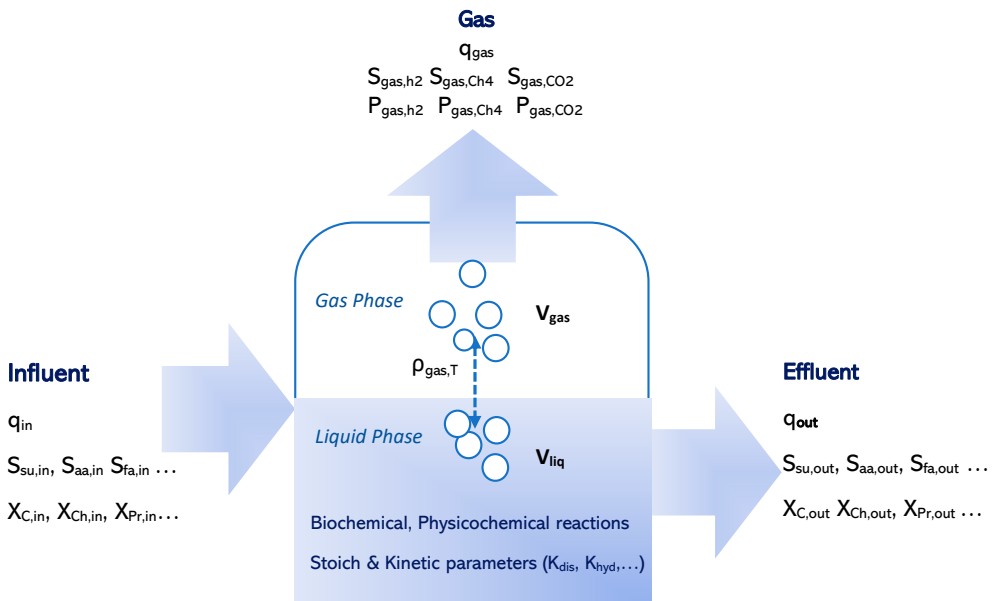

**Figure 2.** Mass balance boundary around CSTR in the anaerobic digestion system, adapted from Batstone et al. [7].

Mass balance equations govern the particulate and soluble component concentrations in the liquid phase. According to Batstone et al. [7], a description of each component is required to implement the model and complete mass balances for all model components. As a result, ADM1 grouped these particulates into a state variable termed composite particulates ($X_c$) [33]. Carbohydrates ($X_{Ch}$), lipids ($X_L$), proteins ($X_{pr}$), soluble inerts ($S_1$), and particulate inerts ($X_I$) are also defined as disintegration particulates, as shown in Figure 2.

Provided that one COD mass unit of $X_C$ disintegrates, the following results will be obtained:

$$\begin{aligned} f_{S_I\,X_C}S_I + f_{X_I\,X_C}X_I + f_{ch\,X_C}X_{ch} + f_{pr\,X_C}X_{pr} + f_{li\,X_C}X_{li} \\ = 0.1S_I + 0.25X_I + 0.2X_{ch} + 0.2X_{pr} + 0.25X_{li} \end{aligned} \quad (1)$$

The COD balance matters as much as the sum of all $f_{i;xc} = 1$. However, it is proposed that the nitrogen concentration of $X_C$ ($N_{xc}$) be 0.002 kmol N/kg COD. Instead, the nitrogen concentration is 0.0021(kmol N) based on the disintegration output and parameter values from Batstone et al. [7]. Thus, it indicates that 0.1 mole of nitrogen is produced for every kilogram of COD that degrades, 5% more than the original. Although the "default" parameter values should automatically close the mass balances, the nitrogen content and production from composites are likely to vary and may need to be adjusted for each specific case study [8].

### 2.4.2. Mathematical Equations

For each component considered in the liquid phase and particulates, the mass balance equations can be written in the general form of Equations (2) and (3).

$$\frac{d}{dt}\left(S_{liq,i}\right) = \frac{Q}{V_{liq}}\left(S_{in,i} - S_{liq,i}\right) + \sum_{j=1}^{19} \rho_j v_{i,j} \quad (2)$$

$$\frac{d}{dt}\left(X_{liq,i}\right) = \frac{Q}{V_{liq}}\left(X_{in,i} - X_{liq,i}\right) + \sum_{j=1}^{11} \rho_j v_{i,j} \quad (3)$$

where $S_{liq,i}$ = Soluble state variables no. 1 to 12, 25, and 26; $X_{liq,i}$ = particulate state variables no. 13–24; Q = influent or feed volumetric flow, equal to outflow (assuming no accumulation); $V_{liq}$ = volume of the digester; $S_{in,i}$ = soluble component concentration,

influent stream; $X_{in,i}$ = concentration of influent stream, particulate components; $\rho_j$ = specific kinetic rates, calculated as:

$$\rho_j = K_m \left( \frac{S}{Ks + S} \right) X \tag{4}$$

where $K_M$ = Monod kinetic rate; $K_S$ = substrate concentration.

Gas Calculations

The rate equations for the gas phase are quite similar to the rate equations for the liquid phase. In ADM1, mass balance for the gas phase components considers methane, hydrogen, and carbon dioxide, with the assumption of no heat transfer in the influent flow. Anaerobic digestion system has continuous gas volume; thus, Equations (5) and (6) represent the remaining three state variables, 27 to 29.

$$\frac{d}{dt} \left( S_{gas,i} \right) = \frac{q_{gas}}{V_{gas}} \left( S_{gas,i} \right) + \frac{V_{liq}}{V_{gas}} \rho_{T,i} \tag{5}$$

where $S_{gas,i}$ = gas phase variables no. 27–29; $q_{gas}$ = total gas outflow; $V_{gas}$ = volume of gas occupying the digester headspace; $\rho_{T,i}$ = specific mass transfer rate of gas.

$$\rho_{T,i} = K_{La} \left( K_H S_{gas,i} - S_{liq,i} \right) \tag{6}$$

where $k_{La}$ = mass transfer coefficient in gas–liquid volume; $K_H$ = Henry's law coefficient.

The gas law can be used to calculate the pressure of each gas, with all gases being interpreted as an ideal gas at the equivalent temperature with the liquid phase components. As a result, each gas component's partial pressure is governed by the ideal gas law, as shown in Equations (7)–(9). The denominators 16 and 24 are COD equivalents for unit conversion.

$$P_{gas,H_2} = S_{gas,H_2} \cdot \frac{RT_{gas}}{16} \tag{7}$$

$$P_{gas,CH_4} = S_{gas,CH_4} \cdot \frac{RT_{gas}}{64} \tag{8}$$

$$P_{gas,CO_2} = S_{gas,CO_2} \tag{9}$$

Assuming that water vapor has saturated the reactor headspace.

Too often, the headspace is saturated with water vapor; therefore, the maximum headspace pressure is determined by the water vapor ($P_{gas}$) along with $CH_4$, $CO_2$, and $H_2$. Thus, its partial pressure ($P_{gas,\ H2O}$) needs to be subtracted from $P_{gas}$. Hence, the gas production rate, $q_{gas}$, can be calculated using Equation (10).

$$q_{gas} = \frac{R \cdot T}{P_{gas} - P_{gas,H_2O}} V_{liq} \left( \frac{\rho_{T,H_2}}{16} + \frac{\rho_{T,CH_4}}{64} + \rho_{CO_2} \right) \tag{10}$$

Equation (11) is used to modify the benchmark water vapor pressure at 25 °C to the apparent temperature because water vapor pressure is significantly reliant on temperature (T).

$$P_{gas,H_2O} = 0.0313 \cdot e^{\left( 5290 \left( \frac{1}{298} \right) + \frac{1}{T} \right)} \tag{11}$$

pH Calculations

$H^+$ ion calculations are performed using the "pH solver" found in Rosén and Jepsson [34]; this is because Batstone et al. [7] are not particularly precise on how to compute pH for the DE-implementation. As recommended by Rosén et al. [34], acid concentrations were determined by summing up the number of acid-base pairs and the negative concen-

trations of base in an acid-base pair. The differential equation for determining the amount of negatively charged particles in an acid-base pair is shown in Equation (12). According to Rosén et al. [34], the pH inhibitory impact is also considered (see Table 2).

$$\frac{dS_{acid,i}}{dt} = -K_{A,B\ ;i}(S_{acid-,i}(Ka_{acid} + S_{H^+}) - Ka_{acid}S_{acid,total}) \tag{12}$$

where $S_{acid-,i}$ = negatively charged acid pair; acid number I, ion number i; $K_{a,i}$ = acid-base equilibrium coefficient for acid i; $k_{A;B;i}$ = acid-base reaction kinetic parameter for acid i; $S_{H^+}$ = concentration of H$^+$ ions; $S_{acid,total}$ = acid-base pair sum; $Ka_{,acid}$ = acid-base equilibrium co-efficient.

**Table 2.** Inhibition factors and equations summarized.

| Type of Inhibition | Description | Equation | Affected Process |
|---|---|---|---|
| pH Inhibition | pH inhibition at both low & high pH | $I_{pH} = \frac{1 + 2 \times 10^{0.5(pH_{LL}-pH_{UL})}}{1 + 10^{(pH-pH_{UL})} + 10^{(pH_{LL}-pH)}}$ | All substrate uptake |
| | pH inhibition at low pH only | $I_{pH} = \exp\left(-3\left(\frac{pH-pH_{UL}}{pH_{UL}-pH_{LL}}\right)^2\right)\ \|_{pH<pH_{LL}}$ <br> $I_{pH} = 1\ \|_{pH>pH_{LL}}$ | |
| Competitive Inhibition | Valerate & Butyrate competes for C4 | $I_1 = \frac{1}{1 + S_1/S}$ | Butyrate, valerate, C4 uptake |
| Non-competitive Inhibition | Hydrogen and free ammonia inhibition | $I_{h2} = \frac{1}{1 + S_{h2}/K_{I,h2}}$     $I_{Nh3} = \frac{1}{1 + S_{Nh3}/K_{I,Nh3}}$ | LCFA, Acetate, Butyrate, valerate, propionate uptake |
| Secondary substrate | Inhibition due to limited inorganic nitrogen | $I_{IN,lim} = \frac{1}{1 + K_{S,IN}/S_{IN}}$ | All substrate uptake |

Inhibition

Toxicity and inhibition may have an impact on the degradation process. The incremental loss effect noticed on the kinetic rates can be explained by using the applicable inhibition factors ($I_1$, $I_2$, ..., $I_n$) in conjunction with the specific kinetic rates stated in Equation (4). The model's inhibition factors are the result of combining one or more of the inhibition equations listed in Table 2. Even though the default version of ADM1 only includes four types of inhibition, it is thought to be sufficient for treating common substrates.

2.4.3. Model Assumptions

A simplified model based on the scientific and mathematic fundamentals of mass balance was constructed in this work. For the anaerobic treatment unit's kinetic model, assumptions were made with caution, and system observations were reported for mass balance calculations around the digester. The first assumption was that sewage sludge from municipal wastewater treatment comprises organic matter in a soluble state and thus does not comprise organic and inorganic suspended particles. The digester is entirely homogenized, and the wastewater is heated to 37 ± 2 °C, with consistent substrate concentration throughout the experiment [24,34].

Only acidogenic microbes are thought to be present in the hydrolysis process, whereas both methanogenic and acidogenic microorganisms are present in the remaining AD reactions [7]. Acidogenic bacteria decompose dissolved organic material to volatile fatty acids and $CO_2$ during hydrolysis. Volatile fatty acids (VFA) are transformed into acetic acid, hydrogen, and carbon dioxide in the second stage, which are the key substrates for methanogenesis, resulting in $CH_4$. The hydrolysis process is primarily responsible for the formation of VFA from sugar monomers. However, it is anticipated that unhydrolyzed glucose is transformed into VFA and methane in the anaerobic digester at the same time [7].

In this study, the microbiology of anaerobic digestion focuses on the entire conversion of organic matter to methane by primarily two bacteria populations. Acetogenic and

methanogenic microorganisms are inhibited by unionized acetic acid, which results in methane generation. As a result of the high VFA concentration expressed as HAc, unionized acetic acid (HAc) inhibits the hydrolysis process. Above 200 g N m$^3$, the major source of nitrogen, ammonium ($NH_4$), also has an inhibitory effect. Because the average influent $NH_4$–N concentration was measured to be 11.9 g Nm$^{-3}$, there should be no ammonium inhibition [34].

Because of the high ion concentration and rising temperature in the digester, it is safe to infer that $CH_4$ solubility is nearly nil. The pollutant level of wastewater was measured in terms of glucose, VFA, and volatile organic compounds (VOCs). The total COD concentration in the influent was subtracted from the VFA, which is a measured system variable equivalent to COD (1 g HAc l$^{-1}$), and the residual COD in the wastewater was converted to glucose equivalent using the formula (1 g glucose equivalent to 1.066 g COD).

### 2.5. Sensitivity Analysis

The model is comprised of a complex set of variables and parameters. It is critical to understand how parameters affect the reliability of results when applying simulation results to real-world implementation in a real plant. The topic that is frequently raised is whether any parameters define the space in which calculations are performed that have a bigger impact on findings than others. If this is the case, studying certain parameters to choose them appropriately is more significant than studying parameters that do not have the same impact on results.

Generated methane gas was selected as an important model result outcome to investigate the model's parameter dependence. The effect of methane production was studied by keeping all parameters constant and gradually increasing them one percent at a time. The definition of the derivative can be used to investigate parameter dependency from those recorded effects in gas production; the parameter is denoted by x in Equation (13).

$$\frac{dF}{dx} = \frac{f(x + \Delta h) - f(x)}{\Delta h} \tag{13}$$

However, to make calculations simple and feasible, Equation (13) is converted to Equation (14) instead of utilizing the usual derivative definition:

$$\frac{dF}{dx} = \frac{F(1.01x) - F(x)}{0.01} \tag{14}$$

Using this method for measuring sensitivity (using ×1.01 instead of $\Delta h$), the design for parameter dependence can be thought of as a simplified Jacobian matrix comprising numerically calculated gas production partial derivatives as a function of each parameter studied. This can be written for the parameters p1, p2, . . . . . ., pn as:

$$J = \left[ \frac{\partial q_{gas,CH_4}}{\partial P_1}, \frac{\partial q_{gas,CH_4}}{\partial P_2} \cdots \cdots \frac{\partial q_{gas,CH_4}}{\partial P_1} \right] \tag{15}$$

While this is a straightforward and easily justified method of determining variable reliance, there can be complexities when comparing different parameter dependents. Because the denominator of the derivative definition is dimensionally dependent, all sensitivities have different components, making comparisons of various parameters impractical. A similar approach is used in the definition, but only the numerator is used. Finite difference is a term used to describe this procedure, and it has the following equation:

$$\text{Change in gas production} = |f(1.01 \cdot x) - f(x)| \tag{16}$$

The parameters examined will influence the outcome of methane gas generation using this method. An important feature of this numerical approach to determining parameter dependency is that it is only valid in a single parameter space position; altering any of

the parameters renders the result invalid and needs a new analysis. This holds true even if the input variables specifying how the model works are modified. These will have an effect on the parameter space in which the calculations are performed. As a result, comparing the outcomes of dependent simulations for different sections of the parameter space is intriguing.

If a parameter dependency evaluation is carried out prior to the enhancement of biogas production, the dependence evaluation must be carried out after the optimization to see if any changes in parameter dependency have occurred. The parameter sensitivity changes when the input variables are modified dynamically. Sensitivity was calculated to determine how much a change in input variables affects outflow, and methods for changing input variables dynamically were developed, the process is detailed in Figure 3.

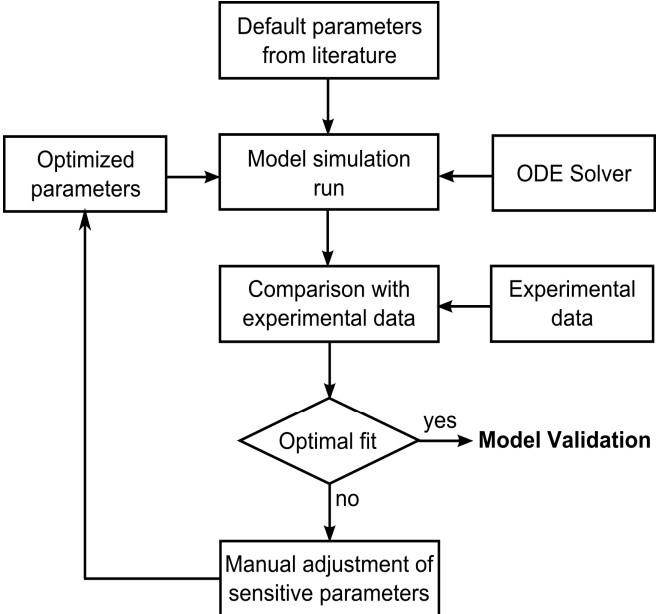

**Figure 3.** Flow chart for running the ADM1 model using Matlab/Simulink$^{TM}$.

Sensitivity Analysis and Variable Dependence of Parameters

Calculated parameter dependence, as previously stated, is valid in a single section of parameter space only. Nevertheless, there are sporadic inflows that change over time. The fluxes entering the digester influence the point in parameter space where computations are conducted. To establish the effects on system sensitivity, multiple analyses, one for each location investigated, are necessary. By gradually changing the inflow while computing the sensitivity of significant parameters, the generalized reliance of sensitivity connected to one variable is investigated. This type of simulation is thus performed using the following method where p is the parameter value, v is the variable value, and f(v,p) is the quantity of $CH_4$ gas created as a function of v and p (measured in $m^3/d$) [35].

### 3. Results and Discussion

A series of simulation tests were performed to evaluate the model implementations offered in this study. These tests include (1) steady-state simulations for comparing transient behavior in detail, (2) two variable simulations for comparing total simulation times, and (3) dynamic simulations for comparing overall simulation times [35,36]. The ODE model is a differential equation model that was used in this work. The simulations were performed on a modern PC with Windows 10 and MATLAB/Simulink R2020a Version 9.8. It is necessary to be familiar with the systems on which the created methods are meant to be used in order to use them. This requires some lab analysis as well as conversion to conventional ADM1 units, which are not always the same as those employed in lab-scale

investigations. As a result, the model can be used to assess and test the compatibility of different raw materials for digestion methods.

A simplified kinetic model based on unstable material balances was used to predict mass balance, VFA, and COD equivalent glucose concentration in a hydrolysis tank, as well as mass balance, VFA, and gas output in a lab-scale anaerobic digester. The algorithm of the kinetic model simulated factors other than COD in the anaerobic tank. Because the internal states of anaerobic digestion cannot be measured using ordinary measurement equipment, the data needed to train and evaluate the pattern recognition systems under review must be generated synthetically. As a result, in order to create the synthetic dataset, a full-scale biogas plant simulation model data is created, and a simulation model is created and calibrated for this biogas plant. Anaerobic digestion modeling using the complex ADM1 has proven to be an effective all-around technique. The ADM1 is often implemented as a non-linear differential equation system [37].

The model that has been implemented can be utilized as a stand-alone model for evaluating a specific system or as the foundation for future applications. This article's framework is meant to be simple to understand and adaptable to new applications. As closely as possible, it follows Rosen and Jeppsson's [36] proposed implementation. Running the model with typical values that lead to steady-state solutions, we obtained the plots in Figures 4–8. Compiling the result to obtain more information from the data gave the plots shown in Figures 9 and 10 for the produced gas and pH, respectively. Gas production is shown as gas flow in Figure 11.

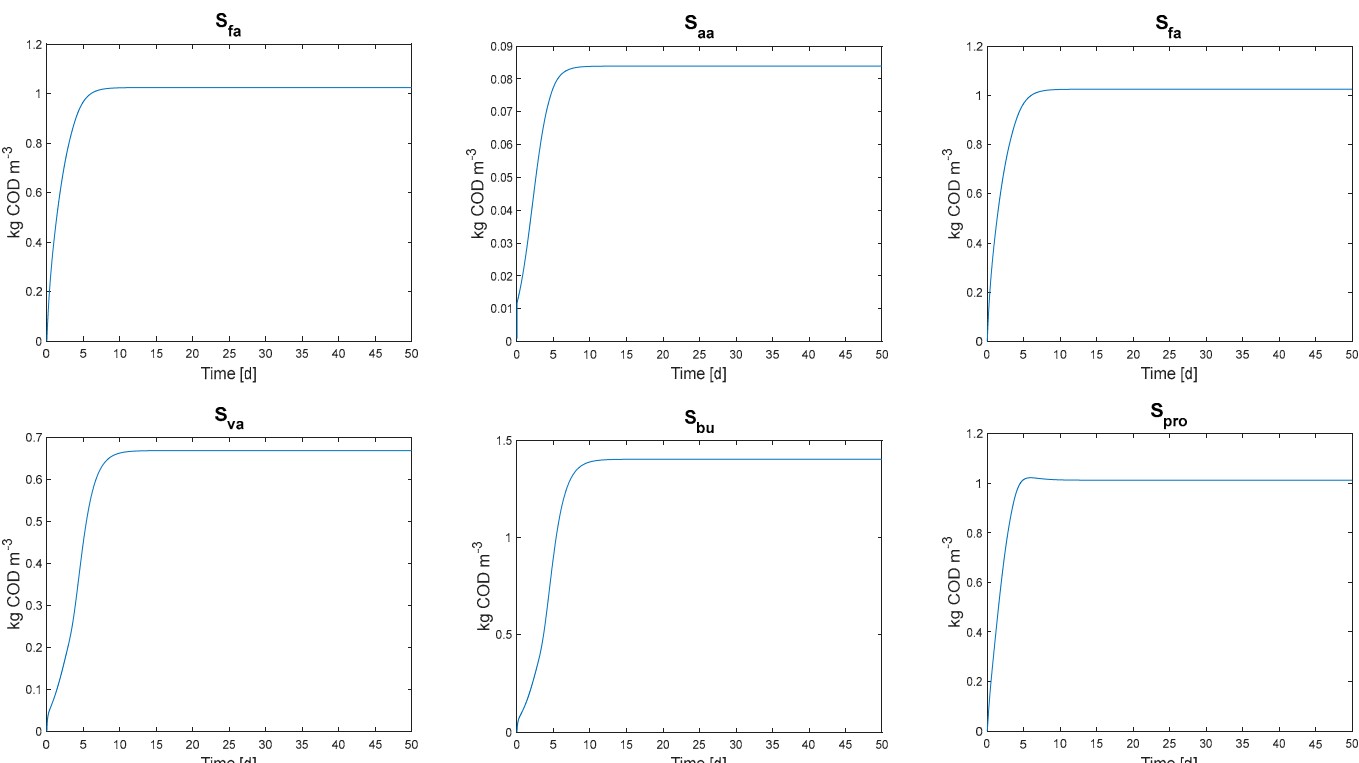

**Figure 4.** Model simulation component output; substrate number output 1–6.

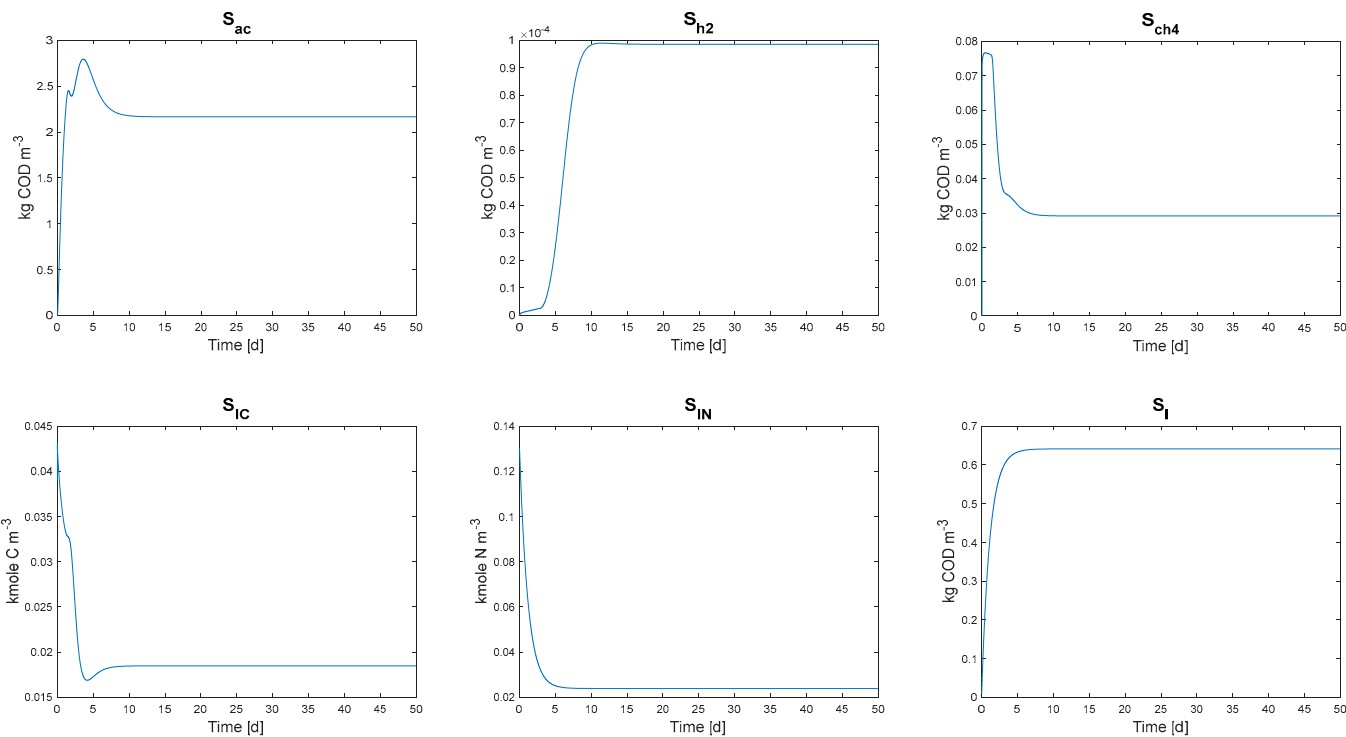

**Figure 5.** Model simulation component output; substrate number output 7–12.

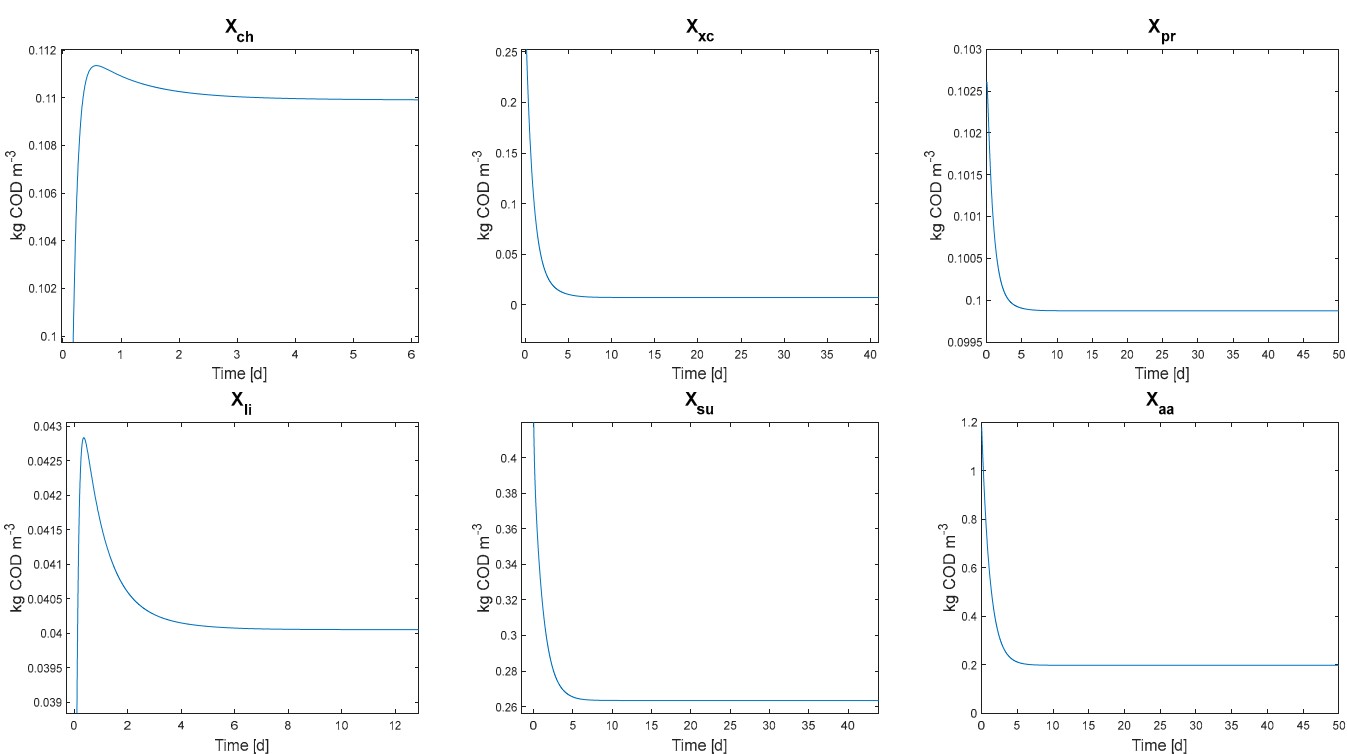

**Figure 6.** Model simulation component output; substrate number output 13–19.

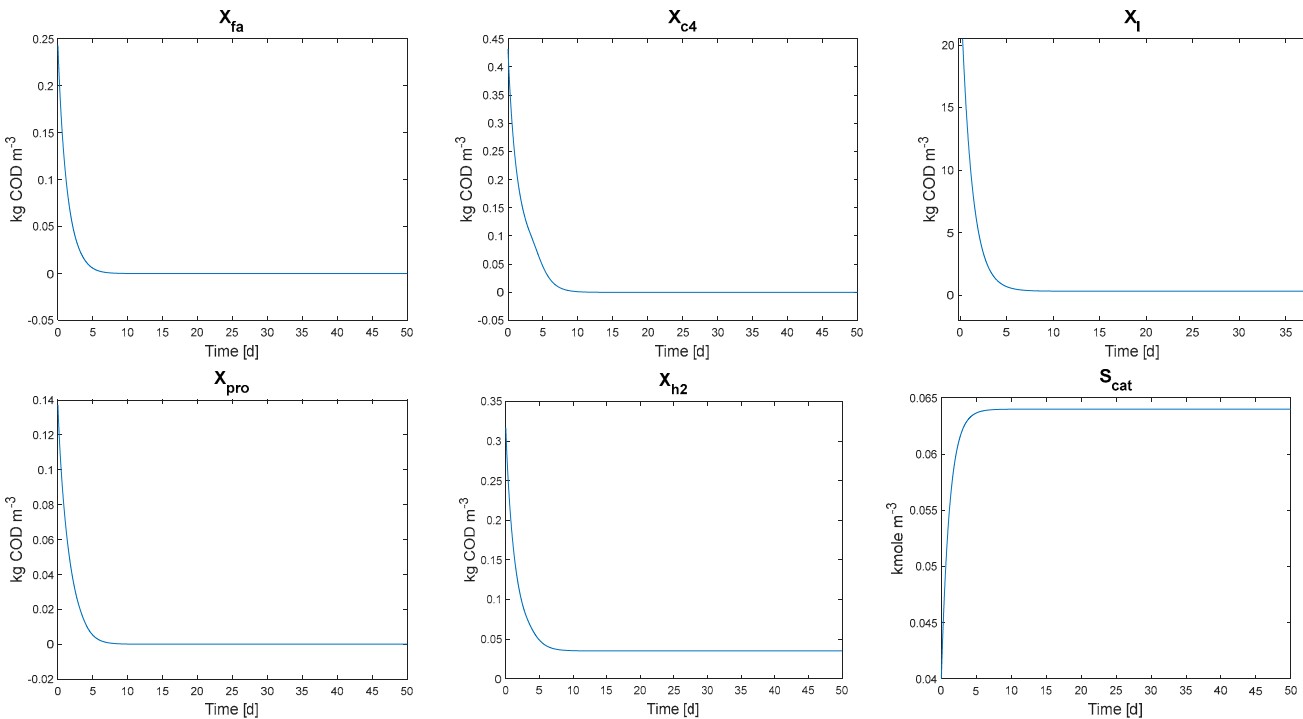

**Figure 7.** Model simulation component output; substrate number output 20–25.

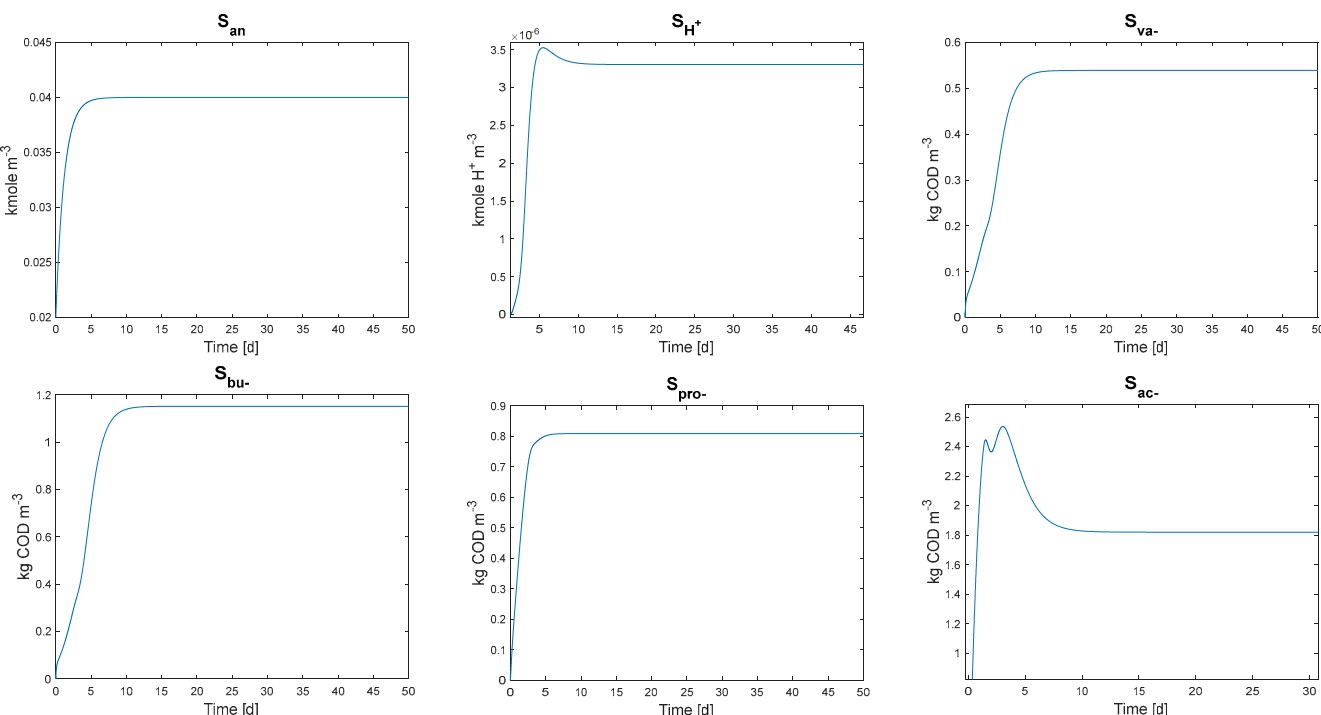

**Figure 8.** Model simulation component output; substrate number output 20–25.

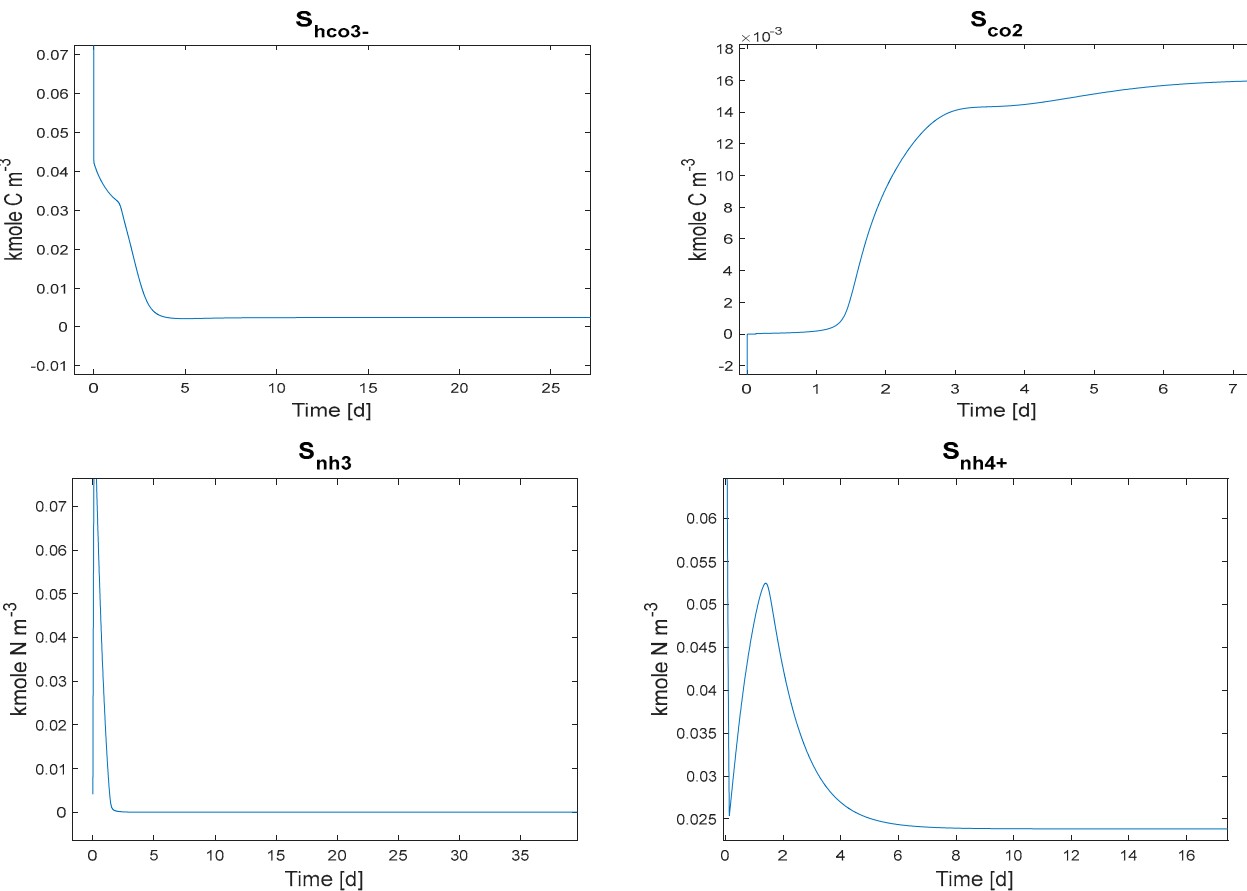

**Figure 9.** Model simulation component output; substrate number output 26–30.

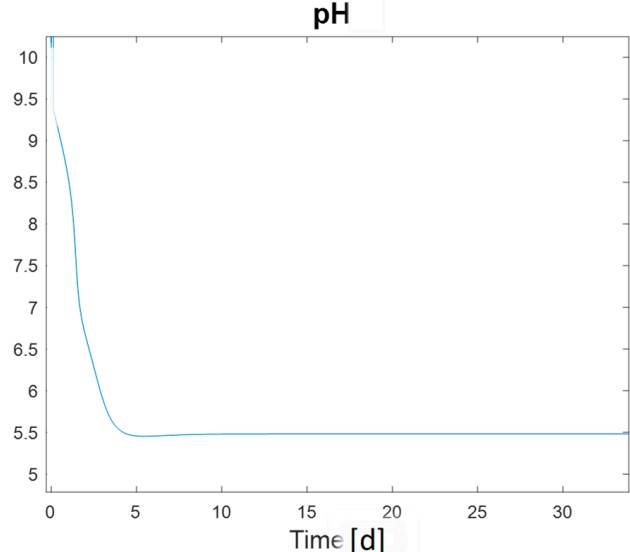

**Figure 10.** Component pH output from model simulation.

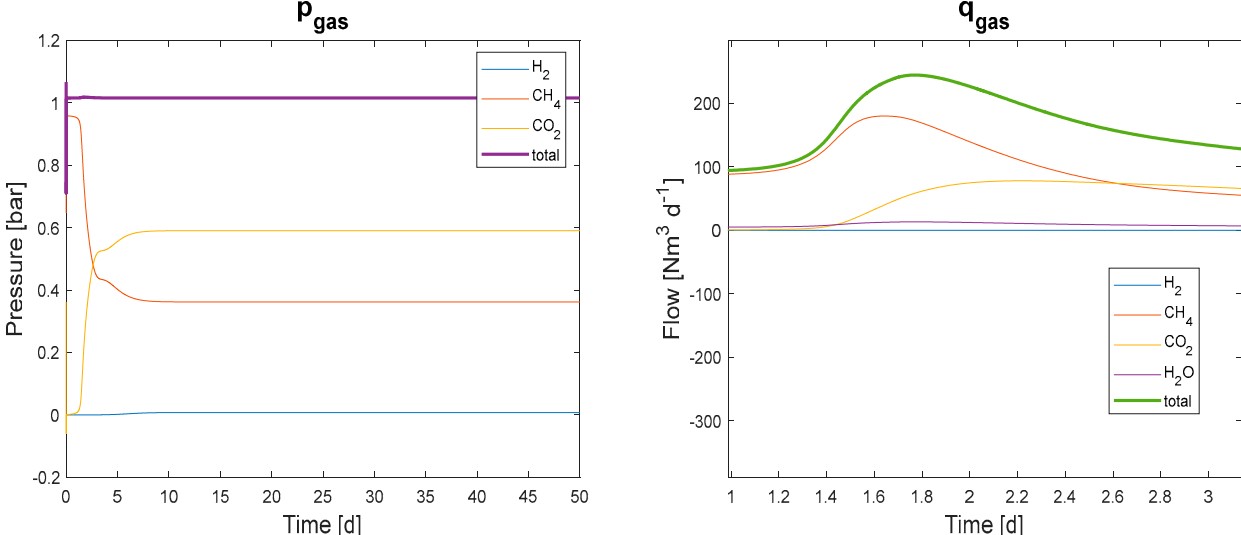

**Figure 11.** Component output for gas production, pressure, and flow rate.

*3.1. ADM1 Simulation Using Measure Parameters*

Figures 4–8 illustrate the models of the six output predictions when multiple independent variables were utilized. This result represents the uncalibrated model. With the exception of ammonia ($S_{IN}$), graphic inspection showed that all simulated outputs closely tracked the pattern of the industrial plant measurements. Because inconsistent responses were reported across various geographical locations and different timescales, the anticipated ammonia trend is considered irregular [38]. This observation could be attributed to the use of ADM1 on an influent with a highly fluctuating substrate composition. Because particulate COD concentration is translated into ADM1 state variables that represent proteins, carbohydrates, and lipids using a constant compositional ratio, disagreement is inherent when the substrate's composition ratio varies [38]. In actuality, as in this instance, the substrate composition can be changed dynamically.

The effect of VSS concentration was greatly understated. Given the projection's close similarity in trend, the lack of correlation is most likely owing to inconsistent kinetic parameters relating to organic particle biodegradation, such as biomass decay, hydrolysis, or biomass growth [39]. As a result, sensitivity analysis, calibration, and cross-validation are likely to improve on these discrepancies. Despite the fact that some parameters were uncalibrated, the pH, methane gas flow ($q_{CH4}$), and VFA fit well. A good pH match is expected since Parra-Orobio et al. [40] pointed out that pH in a well-buffered digester will remain stable. Because the digester has a long hydraulic retention time ((HRT) of $\pm 2$ days), the majority of the alkalinity created during methane production is maintained.

A stable alkalinity buffer is also ensured by the predominantly alkaline substrate. A stoichiometric process converts the three components, i.e., carbon dioxide, acetate, and hydrogen, to methane. As a result, the production rate is proportional to the concentration range of these constituents, which are determined by the kinetics and composition of the influent. To put it another way, COD fractionation into ADM1 system parameters that are calibrated, along with state variables, have a big influence. Because, irrespective of how COD is partitioned, when no inhibition is present, all biodegradability COD will play an active role in methanogenesis activities unrestrictedly, it is possible to derive a relatively accurate estimate of methane gas generation even with uncalibrated parameters. Although the trend was similar, the simulated carbon dioxide flow (qCO2) was continuously higher than the plant data. This is similar to the results obtained by Danielsson [38], in which it was explained that it could be owing to a pH prediction that is lower than the real pH and/or an overestimation of inorganic carbon ($S_{IC}$).

pH is the output that was most precisely modeled. Across a wide range of periods of time and all approaches, including the uncalibrated model, it produced modest residuals. As seen in Figure 10, the used method overstated pH forecasts. Because pH is linked to soluble carbon dioxide concentration, in the model created by the approach, reduced carbon dioxide production explains why pH is higher [40]. The breakdown of monosaccharides, amino acids, and propionate produces carbon dioxide. Nevertheless, as previously stated, the settings were adjusted so that monosaccharide and amino acid metabolism was considerably slowed. The resulting amount of carbon dioxide and propionate produced was lowered [41].

### 3.2. Model Calibration and Validation

Sensitivity Analysis Results

When the model was applied to an existing plant, the sensitivity analysis revealed which components were critical in an actual scenario. The technique advises on which factors are crucial to appropriately select based on their impact on the ultimate result. The sensitivity analysis was carried out using the finite difference approach. The more mathematically rational technique of determining the derivative requires dimensionless units, and because the model was established with dimensional qualities, working with dimensions is easier; hence, the method is dimensional dependent.

The results were produced by modifying each parameter 1% at a time and then calculating the impact on methane gas generation with the newly obtained parameter set. A bar graph was constructed to visualize the data, as shown in Figure 12. If any of the standard parameter set's parameters were comparable to zero, they were retained at 0 during comparison, showing no observable change. Large bars for measured sensitivity were associated with key implementation factors. When the model was applied to a real-world event, those factors became much more important to define precisely [38]. Figure 12 depicts that most parameters are thought to be relevant and that proper measurements of each parameter should be applied in real wastewater plant runs. When working with small volumes, sensitivity is frequently high [37,42]. Low concentrations and quantities were used in this study; hence, the sensitivity was very high. This is expected to decrease when applied to a larger scale, i.e., 2500 $m^3$ WWT plant.

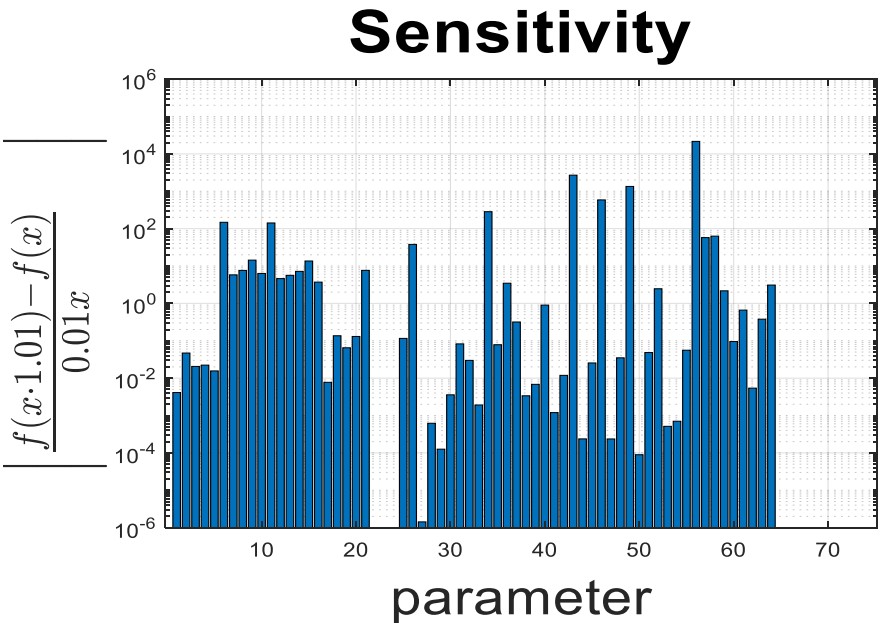

**Figure 12.** Sensitivity analysis plot for the simulated ADM1.

The time required to perform a sensitivity analysis depends on the number of parameters explored, but it usually takes roughly three seconds to evaluate each parameter and three seconds to evaluate the reference. So, the time consumption may be calculated using basic math. The Monte Carlo approach was used to produce a random set of parameters for the evaluation of the model specifications [43,44]. This collection had 500 random parameter pairings. The parameters were set to their default values in order to simulate an IWA-recommended medium-temperature, high-rate reactor. Figure 13 depicts a graph depicting standardized regression, partial correlations, and correlations based on the results obtained for the evaluation of sensitivity analysis.

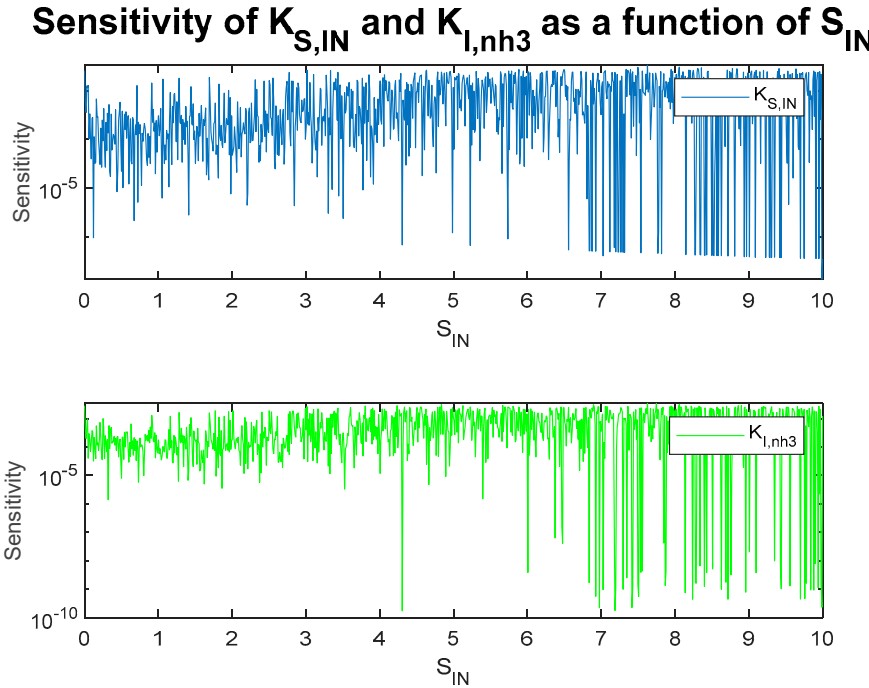

**Figure 13.** Change sensitivity for $K_{S, IN}$ and $K_{i, nh3}$ as a function of $S_{IN}$.

The sensitivity analysis performed is only valid in a subset of the parameter space. The approach for evaluating this produced data is what occurs if the directional movement owing to variations is inflow, as shown it was shown previously by Zhao et al. [45] and Li et al. [46]. This method alters one or more variables while plotting the sensitivity of one or more parameters [47]. Figure 13 shows how the sensitivity of two parameters changes as the amount of inorganic nitrogen in the input changes. To conduct this analysis, the particulate intake was altered from Rosen and Jeppsson's [34] variable C to no particle component input except $X_{xc}$, which was changed to 2.0 kg COD m$^{-3}$.

## 4. Conclusions

The Modified Anaerobic Digestion Model No. 1 (ADM1) is a promising approach for anaerobic co-digestion of sewage sludge for methane production. This model provides a comprehensive understanding of the complex biochemical processes involved in co-digestion systems, enabling more informed decision-making, optimization of operational parameters, and prediction of process outcomes. It offers several key advantages in anaerobic co-digestion, such as:

- Simulating the interactions between substrates, microbial communities, and environmental factors.
- The model accurately represents the dynamics of acidogenesis, acetogenesis, and methanogenesis stages, identifying potential bottlenecks and opportunities for enhancing methane production efficiency.

- It contributes to a deeper understanding of factors influencing digester performance, such as pH, temperature, organic loading rate, and substrate composition. This understanding enables operators to fine-tune operating conditions for optimal methane yield and digester stability.
- The model can facilitate the assessment of the feasibility and economic viability of anaerobic co-digestion projects, guiding investments and resource allocation.
- However, challenges remain in accurate calibration and validation for specific co-digestion systems due to the complexity of microbial interactions and parameter estimation. Further research is needed to refine and expand the model's applicability to various waste streams and co-substrate combinations.

**Author Contributions:** In this research, all authors contributed to the study's conception and design. K.E.M.: Conceptualization, Investigation, Methodology, Writing—original draft, Formal analysis, Validation & visualization and MATLAB simulation. M.O.D.: Conceptualization, Project administration, Writing—review & editing, Supervision S.E.I.: Supervision, Review, and Correction of the Manuscript. T.T.P.: funding acquisition and project administration. All authors have read and agreed to the published version of the manuscript.

**Funding:** This research was partly funded by the Council for Mineral Technology (Mintek) through the Mintek's SET-HCD seed funding project number MCR42415.

**Institutional Review Board Statement:** Not applicable.

**Informed Consent Statement:** Not applicable.

**Data Availability Statement:** Data will be made available on request.

**Acknowledgments:** The authors acknowledge the support of the following organizations: Council for Scientific and Industrial Research (CSIR), University of Johannesburg, University of Pretoria, PEETS, and the University of the Witwatersrand.

**Conflicts of Interest:** The authors declared no conflict of interest.

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
