# Peer review of "Enhancing Methane Production through Anaerobic Co-Digestion of Sewage Sludge: A Modified ADM1 Model Approach"

_fermentation, doi:10.3390/fermentation9090833_

Round 1

Reviewer 1 Report

The manuscript  Application of the Modified ADM1 Model to the Anaerobic Co-Digestion of Sewage Sludge for Methane Production is an interesting topic, and the topic of the manuscript is relevant to the journal targeted. The contents of the manuscript are potentially interesting, however, in my opinion this manuscript need to major revision.

1. There are many paragraphs in the introduction section that need to be integrated.

2.Results section is still too long. This section is merely a big confusion due to lack of coherence and language clarity.

3. There are many paragraphs in the Conclusion section.

4. Conclusion section is not mandatory, it  should be given the specific information about  this research results.

Author Response

Please see the attached response

Reviewer 2 Report

The manuscript Application of the Modified ADM1 Model to the Anaerobic Co-Digestion of Sewage Sludge for Methane Production constructed an ADM1 simulation approach to estimate the performance of an anaerobic digestion of sewage sludge from wastewater and define the biodegradation kinetics. The work examined the validation, calibration and sensitivity of the  developed model. The adjusted ADM1 successfully predicted the the anaerobic digestion process. Actually, there have been plenty of articles about ADM1. Compared to the previous researches, the advantages and distinction of this adjusted ADM1 in this manuscript is not very clear. Besides, the manuscript seems to be incomplete for there is no section 3.1, the number jumped from 2.6.1 to 3.2. Most of the references cited are 4 years ago. Its better to track recent research results.

The language needs careful polish. There are errors in Line 255, Line 268 and Line 277, ect.. 

Author Response

The responses are hereby attached.

Round 2

Reviewer 1 Report

It was improved, I suggest that accept in present form 

Author Response

The authors are grateful to the reviewer for recommending the revised version of the manuscript  for publication.

Reviewer 2 Report

The manuscript showed us a promising approach for anaerobic co-digestion of sewage sludge for methane production. The Modified Anaerobic Digestion Model No. 1 predicted effluent COD, pH, methane, and produced biogas flows with a reasonable degree of accuracy. It also provided a comprehensive understanding of the complex biochemical processes involved in co-digestion systems. There was only a little errors to be revised. Some page numbers of references are missing. (e.g. References 6, 33 and 34). Figure 10 should be redrawn.

Author Response

Reviewer 2: The manuscript showed us a promising approach for anaerobic co-digestion of sewage sludge for methane production. The Modified Anaerobic Digestion Model No. 1 predicted effluent COD, pH, methane, and produced biogas flows with a reasonable degree of accuracy. It also provided a comprehensive understanding of the complex biochemical processes involved in co-digestion systems. There was only a little errors to be revised. Some page numbers of references are missing. (e.g. References 6, 33 and 34). Figure 10 should be redrawn

Response: The authors are grateful to the reviewer for these comments. These comments have been considered in the revised manuscript. For instance, Refs. 6, 33 & 34 have been updated. In addition, Figure 10 has been replaced with one of higher resolution.